# Oral Magnesium Supplementation for Treating Glucose Metabolism Parameters in People with or at Risk of Diabetes: A Systematic Review and Meta-Analysis of Double-Blind Randomized Controlled Trials

**DOI:** 10.3390/nu13114074

**Published:** 2021-11-15

**Authors:** Nicola Veronese, Ligia J. Dominguez, Damiano Pizzol, Jacopo Demurtas, Lee Smith, Mario Barbagallo

**Affiliations:** 1Geriatrics Section, Department of Internal Medicine, University of Palermo, 90127 Palermo, Italy; ligia.dominguez@unipa.it (L.J.D.); mario.barbagallo@unipa.it (M.B.); 2School of Medicine, Kore University of Enna, 94100 Enna, Italy; 3Italian Agency for Development Cooperation-Khartoum, Sudan Street 33, Khartoum 21111, Sudan; damianopizzol8@gmail.com; 4Clinical and Experimental Medicine PhD Program, University of Modena Reggio Emilia, 41124 Modena, Italy; eritrox7@gmail.com; 5Primary Care Department USL Toscana Sud Est-Grosseto, 58100 Grosseto, Italy; 6Centre for Health, Performance, and Wellbeing, Anglia Ruskin University, Cambridge CB1 1PT, UK; lee.smith@aru.ac.uk

**Keywords:** magnesium, diabetes, glucose, meta-analysis

## Abstract

There is a large and growing body of literature focusing on the use of oral magnesium (Mg) supplementation for improving glucose metabolism in people with or at risk of diabetes. We therefore aimed to investigate the effect of oral Mg supplementation on glucose and insulin-sensitivity parameters in participants with diabetes or at high risk of diabetes, compared with a placebo. Several databases were searched investigating the effect of oral Mg supplementation vs placebo in patients with diabetes or conditions at high risk of diabetes. Data were reported as standardized mean differences (SMDs) with their 95% confidence intervals (CIs) using follow-up data of glucose and insulin-sensitivity parameters. Compared with placebo, Mg supplementation reduced fasting plasma glucose in people with diabetes. In people at high risk of diabetes, Mg supplementation significantly improved plasma glucose per se, and after a 2 h oral glucose tolerance test. Furthermore, Mg supplementation demonstrated an improvement in insulin sensitivity markers. In conclusion, Mg supplementation appears to have a beneficial role and improves glucose parameters in people with diabetes. Moreover, our work indicates that Mg supplementation may improve insulin-sensitivity parameters in those at high risk of diabetes.

## 1. Introduction

The literature regarding the health benefits of magnesium (Mg) is exponentially increasing [1]. Recent literature has reported that Mg, involved in more than 300 enzymatic reactions [2], has a wide spectrum of action in cardiovascular [3,4], pregnancy [5], gastrointestinal [6], in infectious [7] and metabolic diseases [8], such as diabetes.

In a previous systematic review and meta-analysis, it was reported that Mg supplementation, compared to placebo, improves several glucose and insulin-sensitivity parameters in people affected by diabetes or have conditions (such as obesity) that put them at increased risk of developing diabetes [9]. In a large meta-analysis including more than 500,000 participants it was found that higher Mg intake is associated with a significant decrease in the incidence of diabetes [10], supported by evidence with a low risk of bias [1]. Unfortunately, the proportion of people meeting the recommended daily allowance (RDA) for Mg is still low. Therefore, in some high risk populations (such as diabetes, older people and pregnant women), oral supplementation is often necessary [11].

Regarding diabetes, several randomized, double-blind controlled trials (RCTs) have investigated the effect of oral Mg supplementation on glucose metabolism parameters. However, these RCTs have an important limitation since they only include a small number of participants [9], as highlighted in a previous systematic review and meta-analysis regarding this topic [12].

Given this background, we conducted a systematic review of RCTs investigating the effect of oral Mg supplementation on glucose and insulin-sensitivity parameters in participants with diabetes or at high risk of diabetes compared with a placebo. It is hypothesized that Mg supplementation would improve glucose and insulin outcomes in those with diabetes and in those with conditions that put them at a high risk of developing diabetes.

## 2. Materials and Methods

This systematic review adhered to the PRISMA statement [13] and followed a pre-planned, but unpublished protocol that can be requested by contacting the corresponding author.

### 2.1. Data Sources and Searches

Two investigators (NV and DP) independently conducted a literature search using several databases including PubMed, EMBASE, SCOPUS, Cochrane Central Register of Controlled Trials and Clinicaltrials.gov without language restriction, from 31 January 2016 until 26 October 2021, including RCTs investigating the effect of oral Mg vs. placebo in patients with diabetes or at high risk of developing diabetes. In a previous meta-analysis of the present group literature on this topic was screened up to 31 January 2016.

In PubMed, the following search strategy was used: (magnesium) AND (‘diabet*’ OR ‘glucose’ OR glycosylated hemoglobin OR ‘insulin’) AND (‘clinical trial’ OR ‘randomized controlled trial’ OR ‘placebo’), adapting the search according to the database. Conference abstracts and reference lists of included articles were hand searched to identify any potential additional relevant articles. Any inconsistencies were resolved by consensus, with a senior author participating in the first meta-analysis on this topic [9].

### 2.2. Study Selection

Inclusion criteria for this meta-analysis were: (i) being an RCT; (ii) double-blind design; (iii) participants with diabetes or subjects at high risk of developing diabetes included; (iv) use of oral magnesium supplementation; (v) assessment of glucose metabolism or insulin sensitivity parameters (see below for further details); (vi) sufficient quality, as assessed by Jadad’s scale [14] ≥ 3/5 points. We considered it diabetes when the diagnosis was made according to the criteria proposed by the American Diabetes Association, i.e.,: fasting plasma glucose (FPG) ≥ 126 mg/dl (=7.0 mmol/L), or 2 h PG ≥ 200 mg/dL (=11.1 mmol/L) during oral glucose tolerance test (OGTT) with 75 g of glucose, or glycosylated hemoglobin (HbA1C ≥ 6.5% (=48 mmol/mol), or random PG ≥ 200 mg/dL (=11.1 mmol/L). [15] Several conditions were considered as placing people at high risk of diabetes, including obesity/overweight, metabolic syndrome, renal failure, family history of diabetes, prediabetes, and others. In the case of disagreement between the two reviewers, a third senior reviewer with expertise in diabetes and metabolic disease (LD) was involved and a consensus among the three reviewers was reached. RCTs were excluded if they: (i) did not include humans; (ii) used a control group taking other substances than placebo; (iii) investigated the effect of Mg supplementation on glucose/insulin-sensitivity parameters in healthy participants.

### 2.3. Data Extraction

Two independent investigators (JD and DP) extracted key data from the included articles in a standardized Excel spread sheet and a third independent investigator (NV) checked these data. For each article, we extracted data on authors, year of publication, country, glucose metabolism/insulin-sensitivity end points, condition, study design (crossover or parallel), medications used for the treatment of diabetes, type of Mg used in the trial with daily dosage, and follow-up duration (in weeks). Moreover, we extracted data on treatment with Mg or placebo, mean age, body mass index (BMI), and number of women at baseline. When information was missing, first and/or corresponding authors of the original article were contacted at least four times to obtain unpublished data.

### 2.4. Outcomes

The primary outcomes were parameters of glucose metabolism and insulin sensitivity. Regarding glucose metabolism, we included FPG, glycosylated hemoglobin (HbA1c), plasma glucose after the 2 h OGTT. Regarding insulin sensitivity, we extracted data on fasting insulin levels or after 2 h OGTT, homeostatic model assessment-insulin resistance (HOMA-IR).

### 2.5. Quality Assessment

Two authors (JD and DP) completed scoring using the Jadad’s scale [14] for assessing the quality of the RCTs included. This quantifies the trial quality based on the description and appropriateness of randomization (2 points), blinding procedures (2 points) and description of withdrawals (1 point). A value of 3 (over a maximum of 5) usually indicates a low-quality study at high risk of bias [16].

### 2.6. Data Synthesis and Analysis

All analyses were performed using STATA version 14.0 (StataCorp). Outcomes with at least two studies were meta-analyzed; outcomes with less than two studies, i.e., not suitable to be included in meta-analyses were excluded.

The primary analysis compared parameters of glucose metabolism or insulin sensitivity between participants treated with oral Mg supplementation vs placebo. We calculated the difference between the means of the treatment and placebo groups using follow-up data through standardized mean differences (SMD) with their 95% confidence intervals (CIs), applying a random-effect model [17]. Heterogeneity across studies was assessed by the I^2^ metric and χ^2^ statistics. In case of significant heterogeneity (I^2^ ≥ 50%, *p* < 0.05) and for outcomes having at least four studies, we conducted a series of meta-regression analyses using as moderators the continent in which the study was performed (categorized as America vs. others), follow-up length, differences in baseline mean age, BMI and mean outcome investigated between Mg and the placebo group [18]. For analyses regarding conditions that included people at high risk of diabetes, the condition investigated (categorized as overweight vs. others) was also explored as a possible mediator of heterogeneity.

As most of the studies reported data on plasma/serum Mg, this parameter was considered when running meta-regression analysis to examine if differences in values of Mg at follow-up between treated and placebo groups could affect the glucose metabolism or insulin-sensitivity parameters at follow-up, independently from heterogeneity.

Publication bias was assessed by visually inspecting funnel plots and using the Begg–Mazumdar Kendall tau [19] and the Egger bias test [18]. Then, to account for publication bias, the trim-and-fill method was employed, based on the assumption that the effect sizes of all the studies are normally distributed around the center of a funnel plot; in the event of asymmetries, adjustment for the potential effect of unpublished (trimmed) studies is employed [20].

For all analyses, a *p*-value less than 0.05 was considered statistically significant.

## 3. Results

### 3.1. Search Results

As shown in Figure 1, among 1606 records initially screened, 48 were retrieved as full texts. Compared to the previous meta-analysis regarding this topic, nine new RCTs were available (three in people with diabetes and six at high risk of diabetes), but one RCT, published in 2019 in diabetic patients, was retracted [21]. Moreover, another study published in 2015 in gestational diabetes was retracted [22], leaving 25 RCTs eligible for the present systematic review and meta-analysis [23,24,25,26,27,28,29,30,31,32,33,34,35,36,37,38,39,40,41,42,43,44,45,46,47].

### 3.2. Study and Patient Characteristics

Full details regarding descriptive findings are reported in Appendix A (for people with diabetes) and Appendix A (high risk of diabetes).

Among the 13 studies including people with diabetes [23,26,31,34,35,36,37,38,39,40,41,42,43], the majority (n = 11) focused on type 2 diabetes, one among pregnant women, one was made among type 1 diabetic people. Ten RCTs had a parallel design, whilst three used a crossover design. The 13 studies followed-up 361 diabetic participants treated with Mg compared to 359 with placebo for a median of 12 weeks (range: 4–48). The participants treated with Mg were on average 50.6 (vs 49.9 in placebo) years, had a mean BMI suggestive of overweight status and were predominantly women. Full details regarding the type of Mg supplementation are reported in Appendix A, with Mg oxide being the most used. The quality of the studies was generally high, as indicated by the Jadad’s scale (Appendix A).

Twelve RCTs [24,25,27,28,29,30,32,33,44,45,46,47] investigated the effect of Mg in conditions that put people at high risk of developing diabetes. Five RCTs included participants being overweight, two among people with metabolic syndrome, two prediabetes (defined as the presence of FPG between 100 and 126 mg/dL or plasma glucose after OGTT between 140 and 200 mg/dL), two in women having polycystic metabolic syndrome, and one metabolically obese with normal weight subjects, fully reported in Appendix A. Eleven RCTs employed a parallel and only one a crossover design.

Altogether these studies followed up 477 participants treated with Mg and 480 with placebo for a median of 14 (range: 4–24) weeks. The participants treated with Mg had a mean age of 42.5 years, a mean BMI of 28.8 kg/m^2^ and were predominantly males; the participants treated with placebo had a mean age of 45.6 years with a mean BMI of 28.9 kg/m^2^. Among the formulations available, similarly to the RCTs included diabetic patients, Mg oxide was the most used form (Appendix A).

### 3.3. Meta-Analysis of the Effect of Magnesium on Glucose and Insulin-Sensitivity Parameters

#### 3.3.1. People with Diabetes

As shown in Table 1, treatment with Mg significantly reduced FPG at follow-up in 325 participants with diabetes compared to 331 taking placebo (n = 11 studies; SMD = −0.426; 95%CI: −0.782 to −0.07; *p* = 0.02), this finding was characterized by a high heterogeneity (I^2^ = 79.0%) (Appendix A). Regarding HbA1c levels, Mg supplementation did not improve this parameter in 301 participants compared to 307 participants taking placebo (n = 10 studies; SMD = −0.134; 95%CI: −0.409 to 0.141; *p* = 0.34; I^2^ = 63.7%). On the contrary, there was no evidence that Mg supplementation was able to improve fasting insulin or HOMA-IR compared to placebo in four studies including 153 people taking Mg and 149 taking placebo.

#### 3.3.2. People at High Risk of Diabetes

Table 1 reports the data regarding the effect of Mg, compared to placebo on glucose and insulin-sensitivity parameters. Overall, Mg supplementation significantly improved FPG in 482 subjects at high risk of diabetes compared to 485 randomized to placebo (11 RCTs; SMD = −0.344; 95%CI: −0.655 to −0.03; *p* < 0.0001; I^2^ = 81.2%) (Appendix A). After trimming 4 studies at the left of the mean, these results remained statistically significant (SMD = −0.565; 95%CI: −0.860 to −0.271).

Similarly, Mg significantly improved 2hOGTT in 3 studies involving 210 participants (SMD = −0.35; 95%CI: −0.62 to −0.07; I^2^ = 0%). Finally, compared to placebo, Mg significantly decreases HOMA-IR in 9 studies (340 Mg vs. 344 placebo) (SMD = −0.234; 95%CI: −0.443 to −0.025; *p* = 0.028; I^2^ = 43.2%) (Appendix A), whilst no effect was observed on HbA1c or serum insulin levels (Table 1).

### 3.4. Meta-Regression Analysis

Figure 2 reports the meta-regression analysis in people at high risk of diabetes using as the exposure the differences in serum Mg between the treated and placebo groups at the follow-up evaluation and the parameters of glucose or insulin-sensitivity metabolism as outcomes. No significant associations were found between differences in serum Mg and differences in FPG in people affected by diabetes.

As shown in Figure 2a, higher differences in serum Mg between the treated and placebo groups at follow-up were associated with higher differences in FPG (beta = −3.50; 95% CI: −5.77 to −1.23; *p* = 0.008; R^2^ = 71.7%) and HOMA-IR (beta = −1.87; 95% CI: −3.69 to −0.13; *p* = 0.04; R^2^ = 100%) (Figure 2b) in participants at high risk of diabetes. On the contrary, as in the previous meta-analysis [9] regarding Mg supplementation on glucose and insulin sensitivity parameters, continent in which the study was performed, follow-up duration, differences in mean age, BMI or in mean outcome parameter at baseline between Mg and placebo treated moderated the present results.

### 3.5. Compliance and Adverse Effects

In trials reporting the number of allocated participants and those finishing the study, we observed that 82% of the allocated participants with diabetes and treated with Mg vs. 75% treated with placebo finished the study, without any difference between the groups (*p* = 0.55). Among participants with higher risk of diabetes, this figure was similar (*p* = 0.81).

No severe side effects (for example, death or onset of cardiovascular diseases) emerged in Mg or placebo groups, also considering the newly included RCTs. The most common side effects observed were of a gastrointestinal nature, particularly diarrhea.

## 4. Discussion

In this systematic review and meta-analysis, including 25 placebo-controlled RCTs on diabetes and conditions at high risk of diabetes, Mg supplementation significantly improved FPG levels in diabetic patients. However, the inclusion of new literature in the present updated review on patients at high risk of diabetes allowed for the observation that oral Mg supplementation not only significantly decreased plasma glucose after 2 h OGTT, but also FPG and HOMA-IR. Finally, the present review also suggests that in both people with and at high risk of diabetes, Mg supplementation is well tolerated and without significant adverse effects.

Mg may improve glucose metabolism and insulin sensitivity via several pathways. First, it is known from experimental models that chronic Mg deficiency is associated with impaired post-receptorial function, consequently reducing glucose utilization in cells [48]. Furthermore, Mg may have an action in improving insulin secretion from pancreatic beta-cells [49]. However, the present meta-analysis failed to show any significant effect of Mg compared to placebo in increasing serum insulin levels, whilst the main action of Mg seems to be attributable to a decrease in insulin resistance as shown by the improvements in HOMA-IR, particularly in those at high risk of diabetes [50], indicating that it is likely that Mg acts better when a deposit of insulin is present [9]. Moreover, findings regarding the improvement in insulin sensitivity are confirmed by other experimental results indicating that Mg is able to decrease oxidative stress [51] and inflammatory parameters [52], two main contributors of insulin resistance [53].

In this regard, it was observed in a review summarizing the role and the effect of Mg in women affected by polycystic ovary syndrome that an association between adequate Mg status and improved insulin resistance is likely, but that oral supplementation with Mg is unlikely to improve glucose and insulin resistance parameters in this specific population [54]. These findings were substantially confirmed by the RCTs included in our meta-analysis and investigating the effect of oral Mg supplementation on glucose metabolism in PCOS.

Furthermore, it is known that Mg may have important effects on other parameters closely related to glucose metabolism, such as body composition, general health, and sleep quality. In particular, serum Mg levels can be associated with inflammatory parameters, as cited before and as reported in a nice case control study involving women affected by fibromyalgia [55]. Moreover, Mg is important in muscle metabolism and it is known that muscle is involved in insulin resistance [56]. Therefore, this pleiotropic effect can help to better explain our findings. Finally, Mg could be important in ameliorating dyslipidemia (particularly if given together with selenium) [57], body composition parameters related to adiposity [58] and sleep quality [59] and all these factors (dyslipidemia, obesity and poor sleep quality) can contribute to poor glycemic control.

As shown in a meta-regression analysis, another relevant finding from the present work is that the differences in serum Mg at follow-up between the treatment and placebo groups significantly correlated with improvements in glucose and insulin sensitivity markers people who are diabetic and in people at high risk of diabetes, confirming findings from the previous systematic review on this topic [9]. Unfortunately, due to the limited studies for other more sophisticated measurements of Mg (e.g., urinary excretion during 24 h), we were not able to assess whether these markers correlate with glucose or insulin sensitivity metabolism, indicating the necessity of future studies in this direction. An important point, however, is that a consistent proportion of the heterogeneity found in the present results is explained by the differences between the Mg serum-treated and placebo groups, indicating that RCTs characterized by a higher compliance and able to increase serum levels can improve glucose metabolism parameters.

The findings of the present meta-analysis should be interpreted within its limitations. First, RCTs consisted of small samples, with a short follow-up period, indicating the necessity of large RCTs with a long follow-up period. Second, dosage/types of Mg largely varied across RCTs: these factors may contribute to the high heterogeneity (I^2^ ≥ 50%) observed in almost all outcomes included. Another limitation is that the evidence is mainly limited to type 2 diabetes. Future studies are needed to understand if Mg supplementation has a role in other types of diabetes, such as type 1 diabetes about which only one RCT is available.

## 5. Conclusions

The present systematic review with meta-analysis suggests a beneficial role of Mg supplementation compared to placebo in improving glucose parameters in diabetic patients, as well as glucose and insulin sensitivity markers in subjects at high risk of diabetes. Future larger studies, with a longer follow-up, are needed to further confirm these results.

## Figures and Tables

**Figure 1 nutrients-13-04074-f001:**
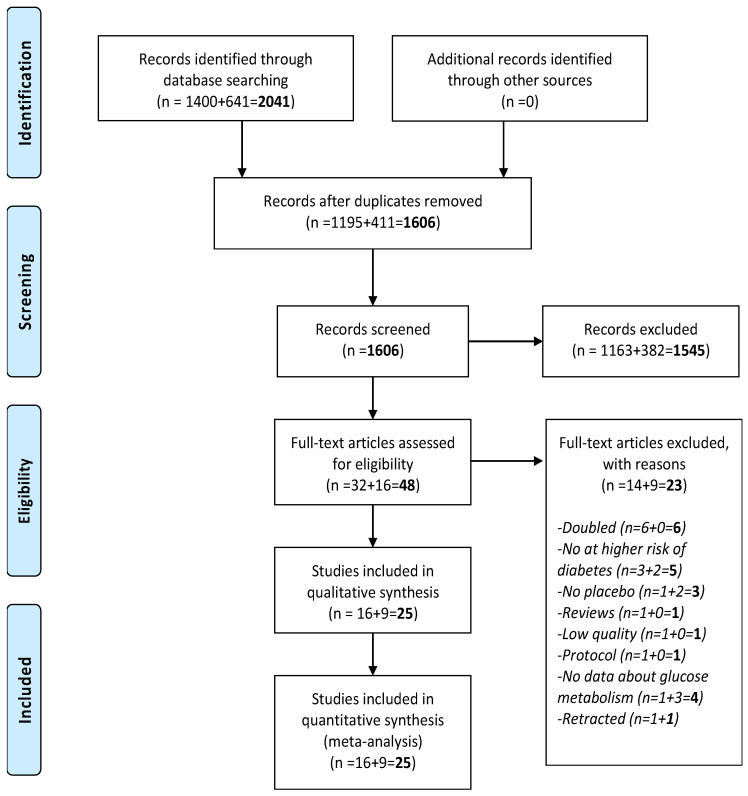
PRISMA flow-chart.

**Figure 2 nutrients-13-04074-f002:**
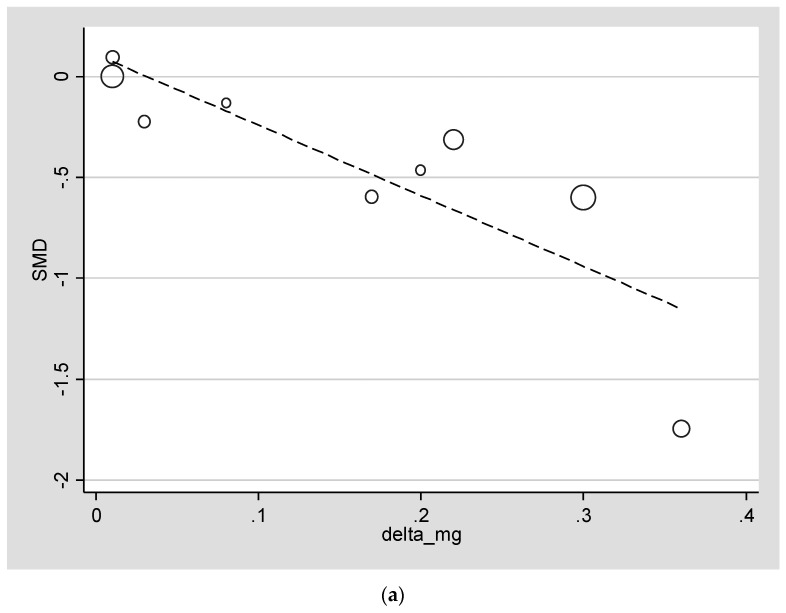
Meta-regression analysis of changes of serum magnesium at follow-up (treated vs. placebo) and (**a**) fasting plasma glucose and(**b**) HOMA-IR.

**Table 1 nutrients-13-04074-t001:** Meta-analysis of eligible studies in diabetes and in conditions at high risk of diabetes.

Diabetes										
Analysis	Number of Studies	Number of Participants		Meta-Analysis			Heterogeneity (I^2^)	Publication Bias		
		Mg	Placebo	SMD	95% CI	*p*-Value		Egger’s Bias and *p*-Value	Trim and Fill (95%CI)	Classic Fail Safe N
**FPG**	11	325	331	−0.426	−0.782; −0.07	0.02	79.0%	−5.84; *p* = 0.02	Unchanged	65
**HbA1c**	10	301	307	−0.134	−0.409; 0.141	0.34	63.7%	5.02; *p* = 0.06	−0.25 (−0.52; 0.03) [2 L]	6
**Insulin**	4	153	149	0.596	−0.576; 1.767	0.32	96.0%	−0.16; *p* = 0.99	Unchanged	4
**HOMA-IR**	4	153	149	−0.169	−0.656; 0.319	0.50	76.9%	−3.98; *p* = 0.63	Unchanged	4
**High risk of diabetes**										
**FPG**	12	482	485	−0.344	−0.655; −0.03	<0.0001	81.2%	1.18; *p* = 0.63	−0.565 (−0.860; −0.271) [4 L]	71
**2hOGTT**	3	105	105	−0.35	−0.62; −0.07	0.01	0%	1.38; *p* = 0.15	−0.41 (−0.64 to −0.18) [2 L]	2
**HbA1c**	2	70	74	−0.275	−1.032; 0.481	0.48	69.3%	Only two studies		
**Insulin**	9	296	296	−0.059	−0.234; 0.116	0.51	11.0%	−2.17; *p* = 0.09	Unchanged	0
**HOMA-IR**	9	340	344	−0.234	−0.443; −0.025	0.028	43.2%	−0.57; *p* = 0.77	Unchanged	15

Abbreviations: CI—confidence intervals; FPG—fasting plasma glucose; HbA1c—glycosilated hemoglobin; HOMA-IR—homeostatic model assessment-insulin resistance; Mg—magnesium; PLC—placebo; SMD—standardized mean difference.

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
