# Peer review of "Oral Magnesium Supplementation for Treating Glucose Metabolism Parameters in People with or at Risk of Diabetes: A Systematic Review and Meta-Analysis of Double-Blind Randomized Controlled Trials"

_nutrients, 2021, doi:10.3390/nu13114074_

Round 1

Reviewer 1 Report

Dear Editors and authors,

I am grateful to revise this article.

It reports an interesting study about the role of magnesium supplementation on glucose metabolism.

Anyway, in my opinion, the Authors should revise extensively their manuscript taking into account the following suggestions.

General comments: English form need to be deeply revised. In several cases, the syntax of the manuscript is not correct.

Major corrections:

  • Introduction:
  • Line 32: In my opinion the sentence “For example..included” is not appropriate at this level of the text;
  • Discussion: the Authors should be deepen in a more extensive manner the putative mechanisms underlying the benefit of Magnesium and/or Magnesium supplementation on glucose metabolism.

Additionally, it could be interesting to report if there are similar data in other categories of patients that could be more prone to insulin resistance, such as those women affecting by PCOS and/or in menopausal transition. (For example, see and discuss Hamilton KP, Zelig R, Parker AR, Haggag A. Insulin Resistance and Serum Magnesium Concentrations among Women with Polycystic Ovary Syndrome. Curr Dev Nutr. 2019 Oct 3;3(11):nzz108. doi: 10.1093/cdn/nzz108. PMID: 31696157; PMCID: PMC6822014. Farsinejad-Marj M, Azadbakht L, Mardanian F, Saneei P, Esmaillzadeh A. Clinical and Metabolic Responses to Magnesium Supplementation in Women with Polycystic Ovary Syndrome. Biol Trace Elem Res. 2020 Aug;196(2):349-358. doi: 10.1007/s12011-019-01923-z. Epub 2020 Jan 20. PMID: 31960275, etc).

Furthermore, it could be adequate to enrich this section by briefly discussing the impact of magnesium supplementation on other metabolisms, body composition, general health and sleep quality in the light of the possible relationship between these fields and the alterations of glucose metabolism (For example, you could discuss and cite: Andretta A, Schieferdecker MEM, Petterle RR, Dos Santos Paiva E, Boguszewski CL. Relations between serum magnesium and calcium levels and body composition and metabolic parameters in women with fibromyalgia. Adv Rheumatol. 2020 Mar 14;60(1):18. doi: 10.1186/s42358-020-0122-4. PMID: 32171334; Capozzi A, Scambia G, Lello S. Calcium, vitamin D, vitamin K2, and magnesium supplementation and skeletal health. Maturitas. 2020 Oct;140:55-63. doi: 10.1016/j.maturitas.2020.05.020. Epub 2020 May 30. PMID: 32972636.; Zhang Q, Qian ZY, Zhou PH, Zhou XL, Zhang DL, He N, Zhang J, Liu YH, Gu Q. Effects of oral selenium and magnesium co-supplementation on lipid metabolism, antioxidative status, histopathological lesions, and related gene expression in rats fed a high-fat diet. Lipids Health Dis. 2018 Jul 21;17(1):165. doi: 10.1186/s12944-018-0815-4. PMID: 30031400; PMCID: PMC6054837. Nielsen FH, Johnson LK, Zeng H. Magnesium supplementation improves indicators of low magnesium status and inflammatory stress in adults older than 51 years with poor quality sleep. Magnes Res. 2010 Dec;23(4):158-68. doi: 10.1684/mrh.2010.0220. Epub 2011 Jan 4. PMID: 21199787).

Taken together, the discussion should be implemented.

Best regards

Author Response

Reviewer 1

Anyway, in my opinion, the Authors should revise extensively their manuscript taking into account the following suggestions.

R: We would sincerely thank the Reviewer 1 for his/her positive comments and for the possibility to send an improved version of our manuscript.

General comments: English form need to be deeply revised. In several cases, the syntax of the manuscript is not correct.

R: The manuscript was carefully revised by an Author of our team that is an English native speaker.

Major corrections:

Introduction:

Line 32: In my opinion the sentence “For example..included” is not appropriate at this level of the text;

R: Thank you for your comment. We have now deleted this sentence since not essential.

Discussion: the Authors should be deepen in a more extensive manner the putative mechanisms underlying the benefit of Magnesium and/or Magnesium supplementation on glucose metabolism.

R: Thank you so much for your comment. We have now added more space in the Discussion section to the putative mechanisms underlying the positive effects of Mg in glucose metabolism.

Additionally, it could be interesting to report if there are similar data in other categories of patients that could be more prone to insulin resistance, such as those women affecting by PCOS and/or in menopausal transition. (For example, see and discuss Hamilton KP, Zelig R, Parker AR, Haggag A. Insulin Resistance and Serum Magnesium Concentrations among Women with Polycystic Ovary Syndrome. Curr Dev Nutr. 2019 Oct 3;3(11):nzz108. doi: 10.1093/cdn/nzz108. PMID: 31696157; PMCID: PMC6822014. Farsinejad-Marj M, Azadbakht L, Mardanian F, Saneei P, Esmaillzadeh A. Clinical and Metabolic Responses to Magnesium Supplementation in Women with Polycystic Ovary Syndrome. Biol Trace Elem Res. 2020 Aug;196(2):349-358. doi: 10.1007/s12011-019-01923-z. Epub 2020 Jan 20. PMID: 31960275, etc).

R: We sincerely thank the Reviewer 1 for indicating these important papers. The paper of Farsinejad-Marj et al. met the eligibility criteria of our meta-analysis: therefore, it was added to the current analyses. Moreover, in order to satisfy your request, a short comment was added to the Discussion section, as follows:

“In this regard, it was observed in a review summarizing the role and the effect of Mg in women affected by polycystic ovary syndrome that an association between adequate Mg status and improved insulin-resistance is likely, but that the oral supplementation with Mg is unlikely to improve glucose and insulin-resistance parameters in this specific population. [55] These findings were substantially confirmed by the RCTs included in our meta-analysis and investigating the effect of oral Mg supplementation on glucose metabolism in PCOS.”

Furthermore, it could be adequate to enrich this section by briefly discussing the impact of magnesium supplementation on other metabolisms, body composition, general health and sleep quality in the light of the possible relationship between these fields and the alterations of glucose metabolism (For example, you could discuss and cite: Andretta A, Schieferdecker MEM, Petterle RR, Dos Santos Paiva E, Boguszewski CL. Relations between serum magnesium and calcium levels and body composition and metabolic parameters in women with fibromyalgia. Adv Rheumatol. 2020 Mar 14;60(1):18. doi: 10.1186/s42358-020-0122-4. PMID: 32171334; Capozzi A, Scambia G, Lello S. Calcium, vitamin D, vitamin K2, and magnesium supplementation and skeletal health. Maturitas. 2020 Oct;140:55-63. doi: 10.1016/j.maturitas.2020.05.020. Epub 2020 May 30. PMID: 32972636.; Zhang Q, Qian ZY, Zhou PH, Zhou XL, Zhang DL, He N, Zhang J, Liu YH, Gu Q. Effects of oral selenium and magnesium co-supplementation on lipid metabolism, antioxidative status, histopathological lesions, and related gene expression in rats fed a high-fat diet. Lipids Health Dis. 2018 Jul 21;17(1):165. doi: 10.1186/s12944-018-0815-4. PMID: 30031400; PMCID: PMC6054837. Nielsen FH, Johnson LK, Zeng H. Magnesium supplementation improves indicators of low magnesium status and inflammatory stress in adults older than 51 years with poor quality sleep. Magnes Res. 2010 Dec;23(4):158-68. doi: 10.1684/mrh.2010.0220. Epub 2011 Jan 4. PMID: 21199787).

R: Thank you so much for this appropriate comment. We have now added in the Discussion section, the following paragraph:

“Furthermore, it is known that Mg may have important effects on other parameters closely related to glucose metabolism such as body composition, general health, and sleep quality. In particular, serum Mg levels can be associated with inflammatory parameters, as cited before and as reported in a nice case control study involving women affected by fibromyalgia. [55] Moreover, Mg is important in muscle metabolism and it is known that muscle is involved in insulin-resistance. [56] Therefore, also this pleiotropic effect can help to better explain our findings. Finally, Mg could be important in ameliorating dyslipidemia (particularly if given together with selenium) [57], body composition pa-rameters related to adiposity [58] and sleep quality [59] and all these factors (dyslipidemia, obesity and poor sleep quality) can contribute to a poor glycemic control.”    

Taken together, the discussion should be implemented.

Reviewer 2 Report

Article ref: Nutrients- 1423787
Title: Effect of Magnesium Supplementation on Glucose Metabolism in People with or at Risk of Diabetes: a Systematic Review and Meta-Analysis of Double-Blind Randomized Controlled Trials
General Comments
The submitted manuscripts reviews on the role of magnesium supplementation on glucose metabolism which is an extension from the previous work (Veronese et al 2016) which includes data from Jan 2016- March 2021 by the same group. The present review would have merit and as mentioned previously includes the recent randomised controlled trials however it will be useful if the title of the review is relooked as the title is exactly the same from the work that was reported previously (Veronese et al, 2016).
Another key issue that the authors needs to rectify is the study inclusions. The authors state that this review includes studies from Jan 2016-March 2021 (Line 64) however the studies presented in Appendix Table 1 and 2 and meta analysis (forest plot 1,2,3 ) of the 25 studies included contains includes 18 studies before Jan 2016 (eg Asemi , 2015, Corica 1994 etc)
Line 128-130. There appears to be missing statement here. This sentence needs revision.
Before any further review could be done to this submitted manuscript the authors needs to rectify and provide correct details

Author Response

Reviewer 2

General Comments

The submitted manuscripts reviews on the role of magnesium supplementation on glucose metabolism which is an extension from the previous work (Veronese et al 2016) which includes data from Jan 2016- March 2021 by the same group. The present review would have merit and as mentioned previously includes the recent randomised controlled trials however it will be useful if the title of the review is relooked as the title is exactly the same from the work that was reported previously (Veronese et al, 2016).

R: Thank you so much for this comment. We have changed the title, as suggested.

Another key issue that the authors needs to rectify is the study inclusions. The authors state that this review includes studies from Jan 2016-March 2021 (Line 64) however the studies presented in Appendix Table 1 and 2 and meta analysis (forest plot 1,2,3 ) of the 25 studies included contains includes 18 studies before Jan 2016 (eg Asemi , 2015, Corica 1994 etc).

R: Sorry for this inconvenience. We have now updated the figures and the tables of the manuscript, including supplementary material.

Line 128-130. There appears to be missing statement here. This sentence needs revision.

R: Thank you so much for this observation. We have checked the sentence and rectified accordingly.

Before any further review could be done to this submitted manuscript the authors needs to rectify and provide correct details

Round 2

Reviewer 1 Report

none

Author Response

none

Reviewer 2 Report

The authors have reviewed and adequately addressed all the comments in the text however there are still errors in the supplementary tables and forest plot figures. These details needs to be corrected to reflect 31st Jan 2016 - Oct 2021. 

Author Response

Thank you for the comments. However, if you double check the forrest plots as well as the supplementary figures, you can observe that studies published between 2016 and 2021 are fully reported. For example, Sadeghian, 2019 is reported in Supplementary S1 and in Figure S1. Similarly for the other studies published between 2016 and 2021. Finally, we removed the studies that were retracted.